# What Should Be Considered When Evaluating the Quality of Home Care? A Survey of Expert Opinions on the Evaluation of the Quality of Home Care in Japan

**DOI:** 10.3390/ijerph19042361

**Published:** 2022-02-18

**Authors:** Seungwon Jeong, Yusuke Inoue, Yasuyuki Arai, Hideki Ohta, Takao Suzuki

**Affiliations:** 1Department of Community Welfare, Niimi University, Niimi 718-8585, Japan; 2Department of Social Science, National Center for Geriatrics and Gerontology, Obu 474-8511, Japan; anant501@obirin.ac.jp; 3Faculty of Health and Welfare Science, Okayama Prefectural University, Soja 719-1197, Japan; y.inoue@fhw.oka-pu.ac.jp; 4Iki-iki Clinic Yuki, Yuki 307-0001, Japan; y_arai@asmss.jp; 5Medical Clinics of ASMss Group, Oyama 323-0014, Japan; h-0101@ceres.ocn.ne.jp; 6Institute for Gerontology, J. F. Oberlin University, Tokyo 194-0294, Japan

**Keywords:** evaluating the quality of home care, home care, quality of care

## Abstract

Intending to obtain scientific evidence to use in developing indicators for evaluating the quality of home care, we surveyed doctors, nurses, and other home care professionals to determine the points they consider to be essential in evaluating home care. We investigated all 901 clinics registered to the National Association of Medical Institutions Supporting Home Care and a random sample of 600 Visiting Nurse Service stations registered to the National Association for Visiting Nurse Service in Japan. A total of 539 questionnaire responses were received (response rate: 35.9%). In this study, a factor analysis revealed four factors to be considered when evaluating the quality of home care: (1) patients’ and family members’ level of satisfaction, (2) home care process, (3) structure of home care, and (4) medical outcomes. The factor of the satisfaction of patients and family members identified in the present study was not considered in previous studies for evaluating the quality of care in Japan. Satisfaction is the point of difference in goals between hospital-based care and home care, and it requires different measurement indicators. Home care professionals expect to help relieve the physical and psychological burden felt by the patient and their family. Thus, on the evaluation indicators of quality of home care, their perspectives from the present study are valuable.

## 1. Introduction

The aging rate in Japan was 28.4% in 2019. As this rate continues to grow, the Japanese population’s medical and long-term care needs are expected to grow even further. In response, the Japanese Ministry of Health, Labour, and Welfare (MHLW) is building a community-based integrated care system to maintain dignity and support independence and is due to be completed by 2025 [1]. In this system, the community provides comprehensive support and services so that older adults can live the rest of their lives independently in the environment in which they are familiar as much as possible. In particular, home care will play an essential role in completing the community-based integrated care system.

In Japan, home care is a medical program for people who cannot commute to hospitals on their own, for example, due to decreased activities of daily living caused by illness, injury, or aging or due to disorders arising from dementia or other mental illnesses. In home care services, various professionals, including doctors, dentists, visiting nurses, pharmacists, dietitians, physical therapists, care managers, and home helpers, collaborate and routinely visit the patient’s home to provide 24-h-a-day treatment and care. Services also include end-of-life care. As a result of an investigation by the Japanese government, over half of people want to spend their last days at home [2]. However, only 14% spend their last days at home, and over 80% spend their end of life in a hospital or other institution [3]. Therefore, we can ask why there is such a discrepancy between the number of people expressing the desire to spend their end of life at home and those who actually choose to. Although home care in the community-based integrated care system is an important part of Japan’s health care delivery system, it has not been implemented as extensively as planned, and the state of how home care should play a role has not yet been thoroughly examined.

The definition and scope of “home care” differ from one country to another. In addition to home care services, long-term care services and short-term care services are also included [4,5,6], depending on each country’s economic situation, political environment, and medical care system [7,8]. Therefore, the services provided for home care vary significantly from country to country and may be understood in a completely different form [7,8,9,10]. Therefore, this study focused on the situation of Japan.

So far, many researchers have been researching the quality of care. For example, medical care in the ordinary healthcare system has evaluated the quality of care based on structure, process, and outcome [11,12,13]. Home care has also been evaluated in a similar framework [6,9,14,15,16]. On the other hand, psychological support to the patient and family [17] and satisfaction with care [18] are also considered in home care. However, much of the research about the quality of home care seems more focused on outcome indicators and the development of case management in the system of care [13,16,18,19].

In the case of home care in Japan, the government gave an additional fee to after-hours emergency assistance and the end-of-life care at home in the medical fee payment of “home care.” Therefore, these indirectly suggest how “home care” should be implemented. However, the field staffs feel that the issues presented above were not exactly focused on the central issues in “home care.” Since there was no basic research on the quality of home care in Japan, we first investigated what factors the experts engaged in in “home care” in Japan to prioritize evaluating the quality of home care. Therefore, in this study, we conducted surveys amongst doctors, nurses, and other home care professionals to determine what we should consider in evaluating the quality of home care.

## 2. Methods

### 2.1. Subjects

We investigated all 901 clinics registered to the National Association of Medical Institutions Supporting Home Care and a random sample of 600 Visiting Nurse Service stations registered to the National Association for Visiting Nurse Service as of July 2017. A postal survey was conducted with self-administered questionnaires (1 August to 31 October 2017). A total of 539 responses were received (response rate: 35.9%), with 394 from clinics (response rate: 43.7%) and 145 from Visiting Nurse Service stations (response rate: 24.2%). The subjects for inclusion in this study were the 532 respondents whose questionnaires had no missing items.

### 2.2. Devising the Questionnaire: Setting Items on the Quality of Home Care

Questions on care quality were extracted from previous studies [2,20,21,22,23,24,25,26,27,28,29,30,31,32,33,34,35,36,37,38,39,40,41].Twenty-three home care specialists (doctors, nurses, physical therapists, occupational therapists, care managers, and social workers) as well as family members of patients were interviewed.A committee of home care experts finalized the items to be used based on previous studies and the interviews. In addition, a pre-test was conducted on working in home care.

### 2.3. Statistical Analysis

In the questionnaire, respondents answered how important they considered each item for assessing the quality of home care. The answers were chosen from a 5-point Likert scale ranging from “not important at all” to “very important”.

Descriptive statistics were performed on items from the questionnaire that home care professionals considered to be important.

To develop indicators for evaluating the quality of home care, a factor analysis was performed to determine which points home care professionals considered to be important for evaluating the quality of home care. The Kaiser–Meyer–Olkin measure of sampling adequacy was examined, and validity was confirmed with the factor analysis. A principal component analysis was used to identify factors, performing a varimax rotation. The factor analysis was repeated, eliminating variables until there were no more variables with a factor loading of less than |0.40| for one factor. SPSS 24.0 (IBM, Chicago, IL, USA) for Windows was used to perform the statistical analysis.

## 3. Results

### 3.1. Characteristics of Respondents

The respondents were 385 doctors (72.4%), 131 nurses (24.6%), and 16 other professionals (3.0%). In all, 69.4% were men, and 30.6% were women. The most common number of years working in home care was 10–19 years (41.6%), followed by less than 10 years (27.7%), 20–29 years (24.1%), and 30 years or more (6.6%) (Table 1).

### 3.2. The Perceptions of Home Care Professionals on Evaluating the Quality of Home Care

A questionnaire survey of 532 home care professionals on the factors related to the quality of home care (max. of 5 points per item) showed the highest points for the following items, in descending order: building rapport with the patient and their family (4.88), multidisciplinary team collaboration (4.82), management of patients’ toileting (4.82), service provider’s ability to communicate with the patient and their family (4.81), determining the family’s level of nursing care fatigue (4.80), collaboration with local hospitals (4.79), and supporting family caregivers (4.76). However, the prognosis of long life (3.22), rate of end-of-life care at home (3.52), and rate of after-hours/holiday emergency assistance (3.54) scored relatively low (Table 2).

The scores from doctors and nurses correlated highly (*r* = 0.9, *p* < 0.001), and there were no differences of opinion between doctors and nurses on any item.

### 3.3. Results of the Factor Analysis on the Quality of Home Care

The factor analysis to examine the components of the quality of home care identified four factors. The Kaiser–Meyer–Olkin measure of indicator validity was 0.92, and the Bartlett’s test of sphericity (*p* < 0.001) confirmed the appropriateness of the factor analysis. In addition, the adjusted Cronbach’s α was at least 0.7 for all factors, indicating that the factors for the evaluation chosen were appropriate. The following are the four factors that were identified (Table 3).

The first factor was named “patients’ and family members’ level of satisfaction” and comprised the level of satisfaction with home care professionals such as doctors and nurses as well as the level of satisfaction of patients and family members with welfare equipment and home care (correlation coefficient among variables α = 0.94).

The second factor was named “home care process” and comprised variables such as support for anxieties about nursing care and daily living arising in the receipt of home care services, the building of rapport with the patient and their family members, and the management of the diet and sleep of patients (α = 0.91).

The third factor was named “structure of home care” and comprised variables such as a framework for information sharing among home care professionals and institutions, a liaison framework, and securing and allocating home care professionals (α = 0.81).

The fourth factor was named “medical outcomes” and comprised variables such as the maintenance and improvement of the independence of patients in daily living, a vital prognosis, and nutritional intake status (α = 0.78).

## 4. Discussion

### 4.1. Variables Considered Important by Home Care Professionals

To develop indicators for evaluating the quality of home care, we surveyed 532 home care professionals to examine which points home care professionals consider to be important for evaluating the quality of home care. We found that doctors and nurses paid particular attention to the satisfaction levels of patients and family members. Conversely, medical outcomes that were the focus for evaluating care quality were considered relatively less considerable.

Our finding that the satisfaction of patients and family members was emphasized more than the medical outcomes may reflect that many recipients of home care are people with diseases or disabilities that are not treatable with modern medical care. For many patients using home care, they and their families may have accepted the disease or disabilities and prioritized living with them rather than treating the medical issues. A few may refrain from choosing treatment due to the invasiveness of the procedure or hospital stays in an unfamiliar environment [42].

Medical professionals who meet a patient’s wishes for treatment build a rapport with the patient and their family and work with them to search for the most optimal treatment method. More than simply supporting the patient to live with their disease or disabilities, medical professionals must also provide support to reduce their patients’ anxieties with regard to treatment and relieve the sense of nursing care burden for family members. In addition to medical professionals, this requires cooperation from nursing care staff, welfare equipment providers, other medical institutions, government entities, and others, and this is why multidisciplinary collaboration and coordination among hospitals and clinics are emphasized. A further value of this study was in extracting insights from a range of medical professionals involved in home care.

### 4.2. Differences between the Existing Evaluation of the Quality of Medical Care and the Evaluation of the Quality of Home Care

Although many studies have examined the quality of care, most have looked at hospital-based care, and research evaluating the quality of home care is lacking in Japan. People generally define the quality of care in hospital care settings as medical outcomes such as an improvement in a vital prognosis or a functional prognosis. In contrast, home care provides healthcare services to people with diseases or disabilities that are not treatable with modern medical care for their remaining life, which is characteristically different from hospital care.

In this study, a factor analysis revealed four factors to consider when evaluating the quality of home care: (1) patients’ and family members’ level of satisfaction, (2) home care process, (3) structure of home care, and (4) medical outcomes.

The framework commonly used to date for evaluating the quality of medical care is the Donabedian model and comprises three categories: structure, process, and outcomes [11,12]. This evaluation model was created primarily for hospital-based medical care. The factors of the satisfaction of patients and family members and collaboration with other specialists identified in the present study are not considered in the Donabedian model for evaluating the quality of care. Satisfaction is the point of the difference in goals between hospital-based care and home care and requires different measurement indicators.

The evaluation of the currently used quality of care and home care evaluation differs in four ways.

First, in home care, family caregivers must provide care at home for patients with medical needs while using services available through the public long-term care insurance system. In addition, home care professionals are expected to help relieve the physical and psychological burden felt by the patient and their family.

Second, hospitals are generally equipped with medical specialists and medical infrastructure and provide complete services. In home care, medical services and nursing care resources in the community must be utilized according to the circumstances of the patients and their family members. To ensure this system works, a higher level of cooperation among institutions and disciplines is required than hospital-based care. Consequently, the relative importance of cooperation is higher among the components of the home care structure.

Third, in home care, more emphasis is placed on the process and structure components overall than on outcome indicator-related components. Various aspects are considered in addition to the medical outcomes, such as the quality of life, in terms of how well the patient’s desired care is achieved, and nursing, in terms of whether there is an adequate system for providing the patient with suitable nursing care services. This may be why the evaluation of the quality of home care requires different indicators than the ones used in evaluating the quality of traditional medical care, which focuses on medical outcomes as the premise.

Lastly, a higher priority must be assigned to satisfaction with home care due to multidisciplinary cooperation than to satisfaction with individual specialists.

### 4.3. Preparing for the Development of the Quality of Home Care Evaluation Indicators

In this study, the indicators for evaluating the quality of home care based on a survey of opinions of home care professionals were verified. Therefore, we can assume that it also verified the content relevance of the indicator components. The next step will be to use the indicators for evaluating home care quality described in this study to collect data on actual activities being conducted, conduct tests on construct validity, and develop a scale.

### 4.4. Limitations

In this study, we surveyed home care professionals to find indicators for evaluating the quality of home care. However, since this study was considered only under the Japanese situation, it is necessary to generalize the health care provider system to other countries. In the future, it is necessary to build a more specific theoretical framework from this study and to observe whether the consciousness of home care professionals and the quality of home care in the actual field are consistent. In this study, the analysis was carried out by quantitative research. Still, it will be necessary to conduct a more thoughtful reflection on the quality of home care through mixed methods, including qualitative research, in the future. In addition, it will be necessary to consider what role “home care” should perform in a difficult situation like the COVID-19 pandemic.

## 5. Conclusions

The demand for home care in Japan, becoming a super-aging society, is expected to grow. However, what should be considered when evaluating home care has not been thoroughly examined.

The present study investigated the viewpoint of home care professionals in evaluating the quality of home care. Furthermore, it laid the groundwork for suggesting the future direction of home care in Japan. Therefore, when evaluating the quality of home care, we recommend that the methods be assessed with various factors and holistic frameworks.

Structure, Process and Outcome, the framework for evaluating quality of care, has been used for various fields of medicine. The present study results suggest that home care professionals need to understand the daily life, anxieties, and the burden of care of patients and their families in addition to the framework for evaluation tools we have been using so far. Particularly, there is a need to pay attention to the satisfaction levels of patients and family members.

## Data Availability

The data presented in this study are available on request from the corresponding author. (The data are not publicly available due to no provision to provide data to a third party in the contents of the ethical review.)

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
