# Peer review of "What Should Be Considered When Evaluating the Quality of Home Care? A Survey of Expert Opinions on the Evaluation of the Quality of Home Care in Japan"

_ijerph, 2022, doi:10.3390/ijerph19042361_

Round 1

Reviewer 1 Report

The manuscript seems to be interesting. However, English editing throughout the manuscript must be required. 

Author Response

We are grateful to the reviewers for their constructive comments and valuable suggestions, which helped us considerably improve our paper. As indicated in the following responses, we have taken all reviewers' comments and suggestions into account in the revised version of our paper. This manuscript received a language editing service.

I appreciate your consideration.

Reviewer 2 Report

The authors carried out a survey among home care professionals (above all doctors and nurses) to determine the most important indicators of home care quality. The main feature of their result is prioritizing patients´ and family members´ satisfaction with the care, which is not so important if evaluating the quality of hospital care.

The paper is well written and can serve as a good background for future investigation and construction of an evaluation tool.

Although the impact of the paper may be reduced due to focusing exclusively on the Japanese situation, it can be published in its current form.

Author Response

We are grateful to the reviewers for their constructive comments and valuable suggestions, which helped us considerably improve our paper. As indicated in the following responses, we have taken all reviewers' comments and suggestions into account in the revised version of our paper. 

I appreciate your consideration.

Reviewer 3 Report

I consider the work to be interesting and relevant, for which I congratulate the authors.  However,

1.- I believe that the introduction is underdeveloped, especially from a theoretical point of view.

2.- As for the instruments used, I believe that it would be richer for the research to complete the surveys with discussion groups, combining quantitative and qualitative analysis.

3.- Finally, I believe that the conclusions could be further developed to give more meaning to the research.

Regards.

Author Response

(The authors gave the same response as above.)

Round 2

Reviewer 1 Report

Thanks for your revision. The revision is sufficient.

Reviewer 3 Report

Thanks to the authors for their efforts.  I consider that the proposed questions have been answered as far as possible.
Best regards.

This manuscript is a resubmission of an earlier submission. The following is a list of the peer review reports and author responses from that submission.